# Revitalization and Branding of Rural Communities in Cameroon Using a Circular Approach for Sustainable Development—A Proposal for the Batibo Municipality

Mudoh Mbah and Anna Franz *

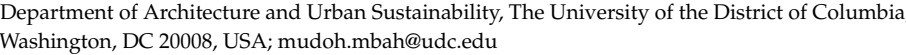

Department of Architecture and Urban Sustainability, The University of the District of Columbia,
Washington, DC 20008, USA; mudoh.mbah@udc.edu
* Correspondence: anna.franz@udc.edu; Tel.: +1-202-657-3794

**Abstract:** Rural communities in Cameroon have high levels of poverty, poor living conditions and lagging sustainable development. Lack of economic, social and physical infrastructure opportunities make these communities unsustainable and impact the quality of life for residents. Existing conditions render these areas unattractive for visitors and external and local investors. Initiatives to reduce poverty and improve living standards have had limited impact to reducing poverty or improving quality of life. The recent signing of Cameroon's decentralization law, giving authority for planning and investments to local council governments now provides an opportunity to rethink existing strategies. Using Batibo, a representative community in the north western region of Cameroon, this paper examines the status of development initiatives and identifies new priorities for planning and steps to improve economic status. Guided by the Theory of Ecological Design and Five Pillars of Economic Development, and using a circular city approach, this paper outlines a concept for town planning and architectural flagship projects that can project the image, culture and heritage of the community and strategies for improving markets. With decentralized governance and re-envisioned priorities, Batibo has an opportunity to become a prototype for sustainable development and model of a quality future in rural Cameroon.

**Keywords:** sustainable development; circularity; revitalization; branding; ecological design; ecological economics; decentralization policy

## 1. Introduction

Cameroon, a country of approximately twenty-five million people, located in central Africa sets as a goal in 2009 to reduce national poverty to less than 10% by 2035 [1]. Data informing the Strategic Plan's primary goal of reducing the country's overall poverty level revealed that between 2001 and 2014, poverty levels in urban areas declined from 17.9% to 8.9% but increased from 51.2% to 56.8% in rural areas [2]. This high poverty rate in rural areas according to the IFAD report [2] is evident in the low level of competitiveness, insufficient infrastructure, high and growing unemployment rate, inadequate planning and coordination capacity, rural exodus, and poor governance. While the government over the past several decades has encouraged selected development initiatives in rural areas and increased rural investment [3], poverty continues to rise, questioning the effectiveness of centrally led strategies and suggesting the need for a new approach if poverty levels are to be significantly reduced by 2035.

Over the past 23 years, the governance structure of Cameroon has been transitioning from a centralized to a decentralized system, and as of 2019, rural communities now have the authority to make decisions about urban planning and development. With recent changes in the governance system and ability to establish local town plans, this paper proposes a circular city approach for sustainable development, which aims at increasing efficiency and effectiveness through the application of targeted actions on a city's assets

and products [4]. In this direction, a master plan for revitalization and branding using architectural flagship projects is identified as essential in catalyzing economic development in rural municipalities.

The purpose of this paper is to demonstrate that circular enabling initiatives such as governance and town planning and their application processes influence the ability of a community to revitalize, become attractive and competitive. Using Batibo, a rural municipality in the northwest region of Cameroon as an example, this paper reviews the legal history leading up to the 2019 law instituting decentralized governance and documents the community's lack of progress in meeting its 2012 strategic vision for development. Applying community-based research, priority initiatives, implementation efforts and the outcomes are examined and reassessed using the circular prioritization chart to determine community priorities for high order value initiatives. Along with road construction, water and electrical infrastructure and vocational education, town planning was identified as a high value initiative. Prior to the 2019 decentralization law, rural communities did not have the authority to develop town plans or make investment decisions that are essential to circularity and necessary for guiding development and improving quality of life. Based on the outcomes of the research and to revitalize and brand the community, a prototype plan including a town center with flagship architectural projects is proposed for consideration by Batibo's local authorities. Planning highlights the importance of creating signature buildings in the town center such as buildings for Cultural History and Center for Innovation and Resilience, to communicate the importance of the community's culture and new ideas for job and market creation, and services that sustain life, such as healthcare, food security and education. In reviewing Batibo's 2012 vision, the paper concludes that while not advanced, the earlier vision was based on a circular approach supporting the Theory of Ecological Design [5–7] and Quality-of-Life factors for economic development [8,9]. With greater autonomy now permitted by the government as specified under the 2019 decentralization law, Batibo's next strategic plan for meeting Cameroon's 2035 goals can re-envision future planning using a circular approach with the new priorities identified in this research.

### 1.1. Legal History and 2019 Decentralization Law

Local councils in Cameroon were created by law no 93/32 of 25 November 1993 and degree no.95/082 of 24 April 1995. This law was followed by an amendment to the 1972 constitution of the country in 1996 that changed the form of the state to a decentralized, unitary state comprising of a central government, regional and local councils [10]. With the local councils now in existence, the legal framework for decentralization prescribed by the 1996 constitution to grant local authorities the administrative and financial independence needed to function separately from the central government did not come until eight years later (that is 2004) when three pieces of legislation relating to decentralization were passed by the national assembly. This was followed by another legislation in 2009 on local fiscal system. As identified by Kofele-Kale [10], the four pieces of legislation included law no. 2004/017 of 22 July 2004 on the Operation of Decentralization, which lays down the general rules applicable to decentralization, law no. 2004/018 of 22 July 2004 to lay down Rules Applicable to Councils, law no. 2004/019 of 22 July 2004 to lay down Rules Applicable to Regions, and law no. 2009/019 of 15 December 2009 on the local fiscal system. Despite the adoption of the law on decentralization, its implementation did not take effect because of processing by the government. This process to transition to a decentralized nation continued until 2019 when another piece of legislation relating to decentralization was signed into law only after political pressure and instability in the English-speaking part of the country seeking greater autonomy and secession from the republic. This means, the entire process to make decentralization a reality in Cameroon as required by the 1996 constitution took twenty-three years. Therefore, between the period of 1996 and 2019, Cameroon was referred as a decentralized state only in theory since the structures to implement it did not exist. As a result, the local councils were still operating under the

centralized system of governance where appointed local representative of the president of the republic supervises the elected officials and make decisions on their behalf regarding the management of their municipality.

One area where the negative impact of the centralized system of governance is most visible is in town planning. Town planning in Cameroon is regulated by law no. 2004/003 of April 2004 [11,12]. It is the law that sets the general rule governing town planning, urban development and building. Chapter 1, Section 4 of the law differentiates between urban centers and communities of at least 2000 inhabitants. According to Section 4.1, human settlements can be classified urban centers only by presidential degree. This is important to note because it provides another reason for the poverty disparity between urban and rural areas, and why rural areas in Cameroon today remained unplanned and under developed. Chapter 1 Section 33 of the law states that town planning master plans can only be drawn in urban centers and groups of councils. This is the same with the development of other town planning documents such as land use plans, sector plans and summary land use plans since they are required to be compatible with the master plan. Areas not classified as urban are not allowed by law to develop master plan, and other town planning documents. The law in Section 8 allows these areas, which, by implication, are the rural areas to follow the general rules governing town planning and building. These rules as laid out in Chapter I are related to stormwater management, building safety, building density, lot occupation, building height and environmental impact assessment. However, as pointed out in several studies based on situation on the ground, in rural areas and in urban centers, the general town planning and building regulations are complex and not or poorly implemented by the local authorities [13–16]. As noted in the above studies, the complex bureaucratic central government structures with overlapping functions in addition to the lack of town planning professionals at the local levels and rural areas in particular, have made application of the law practically impossible. The result of these are communities with no functional town center and disorganized land uses and spatial development. With no authority to address the town planning problems at the local level, improving the quality of life through comprehensive town planning in the rural communities continue to be a challenge.

In 2019, the legal framework for the implementation of the 2004 law on decentralization was finally signed. The law no. 2019/024 of 24 December 2019 instituted the general code of regional and local authorities [17]. In Chapter II, Sections 8–12 [17], it finally devolves to local authorities the powers and appropriate resources needed for their social, economic, cultural, environmental, health, educational, and sports development. One major change identified is that the decentralization law in Section 2 (3) [17] gave all local councils equal status. As confirmed by Kofele-Kale [10], the law abolishes the legal distinction between urban and rural councils. Section 158 of this law for example devolves to local councils Spatial Planning, Regional Development, town planning and housing authority.

*1.2. Background*

Ndenecho [16] in his study of Decentralization and Spatial Rural Development Planning in Cameroon noted that the term rural refers to the communities living in the countryside as opposed to people living in the urban centers. He added that, rural can also imply the people living in an administrative unit of a certain defined size or population. These areas just like most rural areas in developing countries, Ndenecho concluded, are areas with limited services and poor life quality. He described these areas as "depressed regions" [16] (p. 18). Duxbury [18] supported the above definition by stating that rural refers to the population living in towns and municipalities outside the commuting zone of larger urban centers. Just like rural areas in Cameroon discussed by Ndenecho, Duxbury identified problems such as declining and aging populations, problems with youth retention, limited economic and social opportunities for residents, depleting natural resources, loss of local services, and higher costs of living as problems facing rural areas in Canada. Referencing Heimann [19] and others, Meyer [20] defined rural areas as areas that are

sparsely populated, possess limited educational and other community services, and are areas where people farm or depend on natural resources.

In Meyer [20], Kenyon [21] defined rural revitalization as "a process which seeks to reverse rural decline, to develop a more resilient, sustainable and diversified local economy, and to enhance the quality of life of rural communities" [20] (614). Per Meyer, Kenyon highlighted that the positive outcome sort by rural revitalization include stabilizing and increasing the local population, diversifying the economic and employment base, maintaining an acceptable level of service, and preserving special rural attractions. Referring to Vilensiske [22], Ramlee et al. [23] noted that the term revitalization can imply physical, social, cultural and economic dimensions.

Recently, the concepts of circular economy and circular cities has emerged as a new approach to deal with the sustainability challenges facing communities. Regarding circular economy, Girard and Nocca [24] noted that to date, 114 definitions for the term exist. For the United Smart Sustainable Cities (USSC) [4], in one of their several definitions, Mitchel, P. [25] defined circular economy as an alternative to traditional economy (make, use and dispose) in which we keep resources in use for as long as possible, extracting the maximum value from them whilst in use, recovering and reusing products and material. Per the European Environmental Agency [26], circular economy refers mainly to physical and material resource aspects of the economy—it focuses on recycling, limiting and re-using the physical inputs to the economy, and using waste as a resource leading to reduced primary resource consumption. The above definitions are supported by D'Almeida, A.C. [27] who added that in circular economy principles, products can be "made to be made again" [27] (12) through reuse, repair, remanufacture, or recycling. In relation to the circular city, D'Almeida [27] noted that the circular city is the application to urban systems, urban tech, of principles from the circular economics school of thought, which is aimed at building a restorative economy for long term resilience, business and economic opportunities and environmental and social benefits. Key to the concept of circular city according to D'Almeida [27] is the circular city data. He defined circular data as the collection, production, and exchange of data, and business insights, between a series of collaborators around a shared set of inquiries. He added that circular city data is an effort to build a safe environment whereby start-ups, city agencies, and larger firms can collect, produce, access and exchange data, and business insights, through transaction mechanisms that do not necessarily require currency. Circular city program according D'Almeida [27] looks at data knowledge as the energy, flow, and medium of collaboration.

Girard and Nocca [24] also discussed the circular city model extensively. They noted that the circular city model recognizes the importance of the city's systems in analogy to the natural systems (where "nothing is waste"). Typical to the circularity concept per Girard and Nocca is the closure of the loop where linear processes are turned into circular ones. According to them, the circular city is a metaphor for new ways of looking at the city and organizing it. They noted that the idea of circular city is that linear processes can be replaced partly by circular processes and that long-term connections can be established between flows. They concluded that circular approach makes cities independent, rich, and resilient. According to Girard and Nocca [24] and as shown in Figure 1, successful transition towards circular city model requires behavioral changes. They noted that there is a need in changing community life style to implement the circular city model. Though they confirmed that most definitions of circular city focus on material and energy flow, they agreed with Williams' [28] statement that looping actions in cities are related to different themes (with relation to different challenges) that are sociocultural, economic and financial, information, regulatory, political institutions, technical and design and environmental (Figure 1). According to William [28], Girard and Nacco [24] noted that circular-city implementation is an issue related not only to technical questions, but it is related also to a systemic change in society and in restructuring our economy and governance system. They concluded that good governance can reduce or eliminate the barriers hindering implementation of circular initiatives. This view is shared by D'Almeida [27].

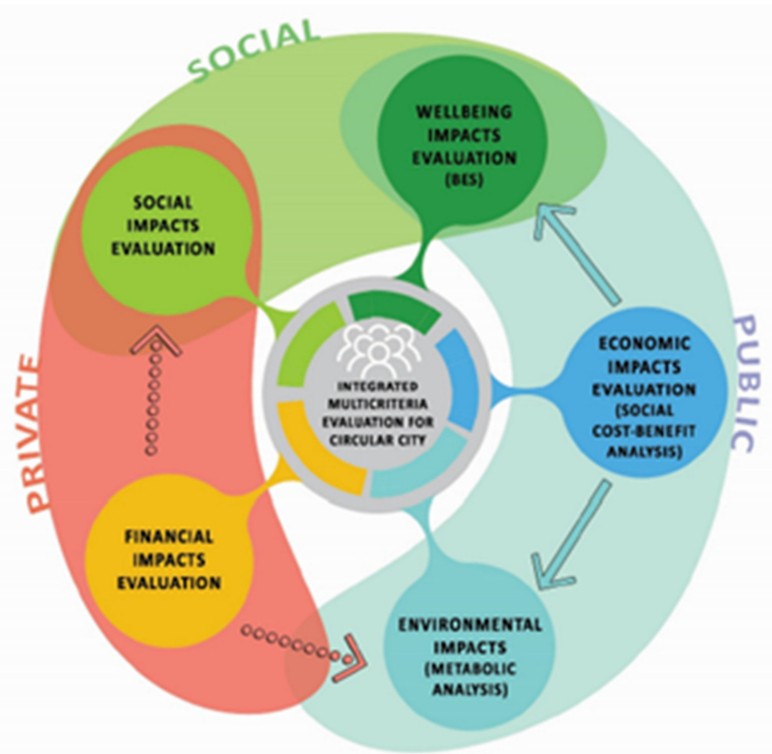

**Figure 1.** The Integrated Evaluation Framework for a Circular City. Reprinted with permission from Girard, L.F., Nocca, F. [24].

In discussing the components of circular city implementation framework, USSC [4] identified city assets and products, circular actions, circular city outputs and circular city enablers as the four components that are involved. They defined circular actions as those specific, discrete, outcome-orientated tasks that can be applied to the city assets and products to improve their utilization and lifespan. They identified sharing, recycling, refurbishing, reusing, replacing and digitizing as examples of circular actions. They defined circular enablers as any entity, activity or initiative that, through its functions, can catalyze and promote circularity in cities. They identified circularity-related strategic planning and policy making, national laws and directives, engaging and ensuring participation of stakeholders, circularity regulations, etc., as examples of circularity enablers. The USSC [4] also identified four steps that are involved in the implementation of circularity. These include assessing the current circularity, determining potential for future circularity and prioritization actions, catalyzing circularity and assessing projected circularity impact. Since communities may not be able to finance the long list of initiatives, USSC [4] identified value and ease of implementation as the two criterion that can be used to prioritize initiatives. They listed economic, social, environmental and cultural impacts as factors determining the value of an initiative. With respect to ease of implementation, they identified cost, time frame and implementation risk (barriers, complexity, health and safety issues) as contributing factors.

Regarding urban planning as a circularity enabler, Girard and Nacco [24] noted that urban planning contributes to trigger flows of energy, materials, services and people to catalyze economic (and not only) development. They concluded that all circular processes and synergies can be implemented in the space of the city/territory through urban planning that represents the institutional tool able to change the existing city organization into a new one based on symbioses and circularization principles. In this direction, they added that urban planning should: promote the conditions of spatial/geographical proximity between resource flows, to achieve synergistic and symbiotic exchanges (i.e., between agricultural production and biological nutrients, compost, bio-gas, etc.) due to geographical-spatial proximity; transforming areas into spaces of centrality, public spaces

of high quality, attractive to people and activities; provide processes for collecting urban meteoric waters and reusing them for non-potable purposes; production of regulations to reduce urban/territorial sprawl; identification or production of particular common assets (libraries, public spaces, places, gardens, monuments, etc.) that can be managed with forms of cooperative/solidarity economy; and promote regulations aimed at stimulating reuse, recovery, repair and maintenance of existing resources, and the use of local materials for new buildings.

Fainstein, Susan [29] defined urban planning as the design and regulation of the uses of space that focus on the physical form, economic functions and social impacts of the urban environment and on the location of different activities within it. Per the American Planning Association [30], the goal of planning is to maximize the health, safety and economic well-being of all people living in our communities. This involves thinking about how we can move around our community, how we can attract and retain thriving businesses, where we want to live and opportunities for recreation. They concluded that planning helps create communities of lasting value. Other terms used in place of "urban" in the definition include town, city, country, regional and rural. According to Carmona, Heath, Oct. and Tiesdell, S. [31], the term urban embraces not only city and town but also village and hamlet. Per Kjell, H. and Nordahl, B. [32], planning in rural areas of European countries has largely been restricted to protecting the country side and agricultural land from built development. They noted that rural planning has often been driven by programs to foster food production, and there seems to be little willingness to change the traditional 'productivist' agricultural approach. On the other hand, Dalal-Clayton, D.B., David. D. and Dubois, O. [33] concluded that rural planning in developing countries is in a state of flux. They noted that the objectives of rural planning have evolved over the years from a focus on increased production, through greater efficiency and effectiveness, to explicit concerns about equity and the reduction of poverty and vulnerability. They added that traditionally, planning in rural areas have been focused on the provision of services such as roads, schools, hospitals, clinics, etc. In urban design, Carmona, Heath, Oct. and Tiesdell, S. [31] noted that the urban center is the single most important idea with which the urban designer works. They emphasized that the center is the centerpiece of 'public realm', the place where major public works, the major public expenditure and the greatest civic art display.

In transitioning towards circular city, Girard and Nocca [24] identified several indicators that can be used to monitor the impacts of projects and initiatives of the circular agenda. These indicators they noted varies from one community to the others. Some of these indicators include; new business opportunities, number of new green jobs, attractiveness in terms of tourist visits, livability, creation of protected green areas, use of renewable resources, amount of waste produced in the city and treated within the city, water efficiency (water issues regarding its treatment and distribution), reuse of building components at the end of life, revenue from recycled goods sold, amount or percentage of waste separation, percentage of incoming/outgoing flows, employment opportunities, spending on waste management, number of individuals trained through the education measures, new collaborations between public agencies and enterprises, social cohesion, population below poverty line, physical and mental health benefits, city attractiveness in terms of creation of recreational and cultural spaces, competitiveness of the economy, etc. Girard and Nocca [24] concluded in their paper that circular cities should not only be attentive to the cycle of waste and resource flow but should place the human being (their health and wellness) at the center of their policies.

In relation to governance identified by Girard and Nacco [24] above, Marume, S.B.M and Jubenkanda, R.R. [34], differentiates between centralization and decentralization governance systems. According to Marume and Jubenkanda [34] centralization means concentration of authority at the top level of the administrative system while decentralization means the dispersal of authority among the lower levels of the administrative system. In a centralized system, they added, the lower levels (called field offices) cannot act on their

own initiative but must rely on the high-level administrators for decision making. This is different from the decentralized systems where the field offices have the authority to make decisions without reference to the headquarters.

### 1.3. Sustainable Planning and Design

Producing good town plans and designs alone without consideration of environmental sustainability is not a good plan in today's world especially for communities that want to transition to circularity. As noted by Williamson, Radford and Bennetts [35] the concept of good has shifted to involve the notion of a building that is sensitive to its environment. This is supported by the Hannover Principles discussed by McDonough and Braungart [36] in Sykes A. Krista [37] when they noted that the standard for designs today are designs that create mutually beneficial relationships between people and the environment. Xing, Jones and Donnison [38] added that to plan and design a sustainable community, natural, social and economic conditions need to be examined holistically. He added that linear approaches can only provide short term economic benefits. To be sustainable, Xing et al. [38] concluded that government policies play very significant role in shaping green actions. Ragheba, Amany et al. [39] used the term green buildings or green architecture to describe the above type of architecture.

### 1.4. Culture Led Development and Branding

The use of architecture flagship projects to revitalize a community is not new. While different features can be used in designing a community's revitalization plan, the use of the culture of the community in the planning and design has been giving special consideration. Groadach and Loukaitou [40] in their paper on cultural development strategies and urban revitalization highlighted the increasingly important role culture plays in the economic development of a community. They noted that for cities to be competitive with others, they have focused in developing cultural activities that can catalyze private development, increase consumption by residents and tourists, improve the image of the city and the quality of life. Gainza, X. [41] concluded that culture have been the pillar for urban regeneration for the past thirty years.

Pastak and Kahrik [42], termed culture-led urban development projects as flagship projects. He defined flagship projects as the development of a building or area with the purpose of giving an impetus for revitalization of its surrounding in physical, economic and social terms. They identified projects such as high-end housing development, luxury shopping centers, tourist attractions and museums, as examples of flagship projects. Though they agreed that cultural flagship projects are good business models for economic development of local communities, they however warned that the impact varies depending on the level of community engagement in the planning phase and the focus of the project. Gainza, X. [41] identified the issue of gentrification as one of the arguments against this approach. Though much emphasis on flagship projects is on architecture, Prilenska [43] noted that flagship projects are not limited to architecture icons alone but to other projects that can bring life into unused spaces such as open air, swimming pool, parks or city beach such as the Frederikshavn Palm beach projects where exotic palm trees were planted.

In relation to the architecture of the flagship projects, the Congress for the New Urbanism [44] in Sykes [37] (p. 62), added that such buildings should have distinctive form because the role they play is different from that of other buildings and places that constitute the fabric of the city. The charter further added that public spaces should be framed by architecture and landscape designs that celebrates local history, ecology and building practice. Moughtin [45] in his book on Urban Design, Street and Square concluded that when public places are designed according to some basic principles and are imbued with a sense of place, they take on an added symbolic meaning.

As noted by Prilenska [43], it is not enough to create attractive image without making it well known locally and internationally. Making a place to be known locally and internationally requires some form of branding. Prilenska [39] concluded that the objective of

branding is to attract inward investment and tourists, and to reinforce local identity and identification of the citizens in their city. Kavaratziz [46,47] also made extensive studies on the role of branding in revitalizing local economy. In his writing, he identified the growing importance of culture and entertainment branding that is implied in cities all over. He also highlighted the effects high profile buildings and other 'landmarks' play in the city's promotion.

### 1.5. Economic Development and Quality of Life

In the discussions of revitalization and circular cities above, one theme common to both is their focused on improving the quality of life of the communities concerned. O'Hara [8,9] identified education, health, environmental quality and recreation, social and cultural amenities and information and transportation access as indicators that can measure the quality of life of a community. The above indicators in addition to others also are identified in the Maine Development Foundation [48] report on indicators of livable communities.

Guiding this paper, the theories applied include the Theory of Ecological Design and the Five Pillars of Economic Development. The Theory of Ecological Design was conceived in 1975 and formalized in a dissertation by Yeang, K.K. 1980 [5]. According to Franz, Sarkani, and Mazzuchi [6] and Franz, A. [7] the theory commenced the industry's vigorous focus on sustainability, shifting the paradigm for standard practice, changing the future course of design and establishing sustainability as a new standard for success. As highlighted by Franz, Sarkani, and Mazzuchi [6,7], Yeang's original model for internal interdependencies, total inputs, total outputs and external interdependencies was later enhanced by red, grey, blue and green ecoinfrastructures (Figure 2). The partitioned matrix for design consideration's systems approach highlights the importance of energy efficient and environmentally sound architecture and the built environments' connection of people, society and culture to nature and water. Whole system design aligning goals, vision, planning, design, construction, project and business management processes with outcomes enhances success, and was applied in this research's methodology and results.

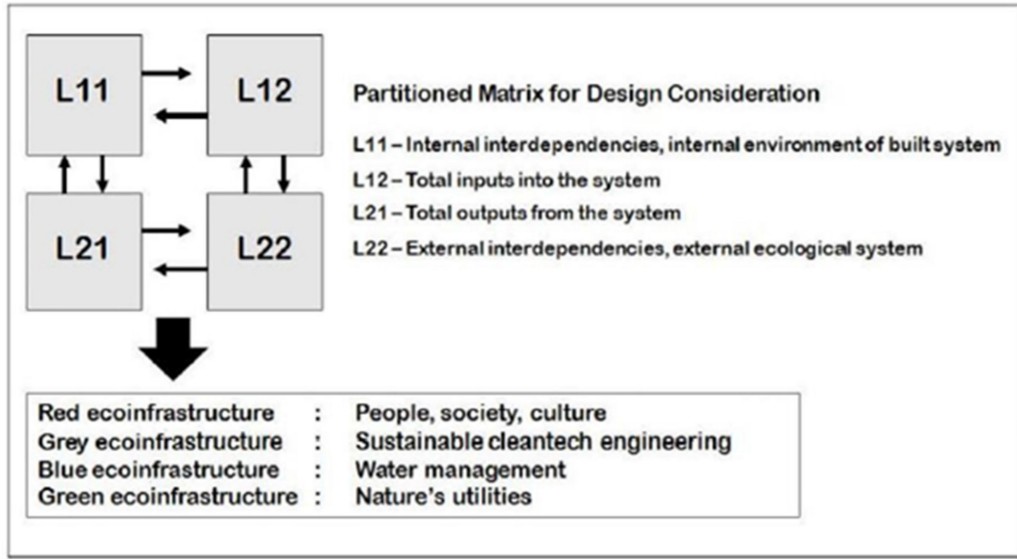

**Figure 2.** Yeang's Enhanced Theory of Ecological Design. Reprinted with permission from © Common Ground Research Networks, All Rights Reserved (permission—cgscholar.com/cg_support) [6].

O'Hara [8,9] in the Five Pillars of Economic Development, a study of sustainable future for Ward 7 and 8 in Washington D.C., explains why some communities develop and others do not, what works in economic development of a community and if economic development can be influenced to make communities prosper. In this study, she criti-

cized the base theory of economic development approach, which argues that a region's economy will grow if the base sector thrives. She highlighted why this traditional model is problematic and offers a new model called the five pillars of economic development. Contrary to the traditional model, her model focuses on tracking and improving those factors that contribute to a high quality of life (QoL) of a community. The model focuses on lead indicators rather than the lag indicator for economic development proposed in the traditional economic development model approach. The five-pillar model proposes (1) health, (2) education, (3) environmental quality and recreation, (4) social and cultural amenities and (5) information and transportation access as the key indicators that can measure the community's likelihood of long-term economic success (Figure 3).

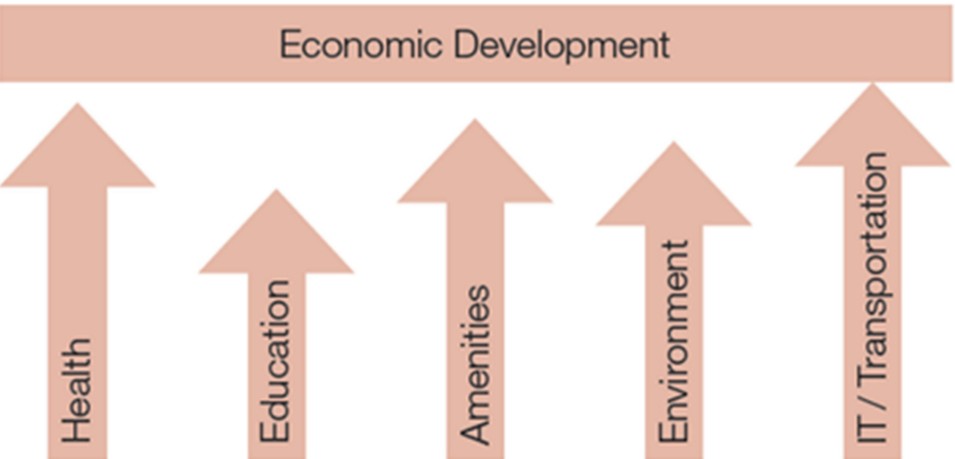

**Figure 3.** The Five Pillars of Economic Development. Reprinted with permission from O'Hara, S. [8].

To buttress her argument for the five-pillar proposal model, O'Hara emphasized that the QoL must include the vision of the community for a quality future. She noted that these indicators are not substitute to for a community's vision for a high QoL, but they can help make the vision concrete. O'Hara concluded that the five pillars of economic development model depends on local information and local knowledge.

This paper builds on the above concepts and the authors' proposed model in Figure 4 depicting the relationship between governance and community revitalization expectations. The model shows that governance is at the center of any change that can occur in a community. Its role operates in a circle whereby the outcome of what comes out of a community, depends on what the government puts into the community.

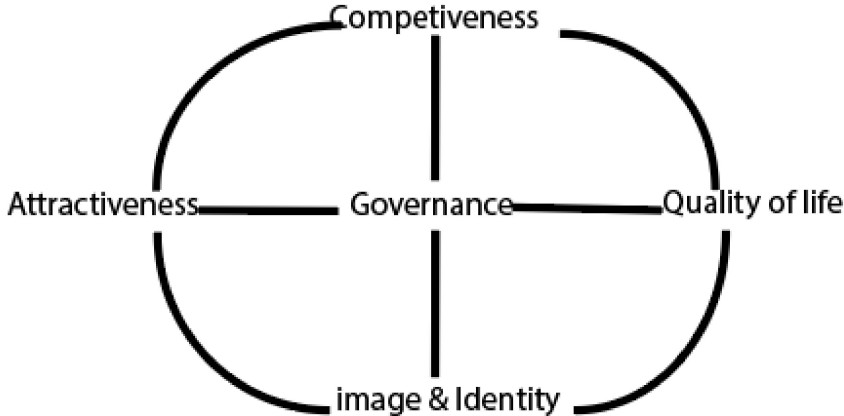

**Figure 4.** Relationship between governance and community development.

## 2. Materials and Methods

Batibo is a sub administrative unit in Momo Division in the North Western region of Cameroon (Figure 5a,b). It is a rural municipality made up of twenty-two villages with Batibo town as the head quarter and seat of the local government administrative authorities. It is located approximately 48 km from the south western corner of the Bamenda metropole, the capital city of the north western region (Figure 6a,b). By 2012, it had a projected population estimate of 74,362 inhabitants [49]. The municipality has a rural council headed by elected officials who manage some activities on the behalf of the people. The subdivisional officer appointed by the president of the republic is the representative of the president of the republic in the area and is responsible for supervising the activities of the local councils. The main economic activity in the area is peasant agriculture, which employs about 90% of the population. The community has no town planning document or land use map to guide development in the area. Culturally, it is a community where music and dancing, traditional religion, food and drinking of local palm wine are very special. The cultural landscape is filled with elements of traditional architecture, arts, crafts and secret forests.

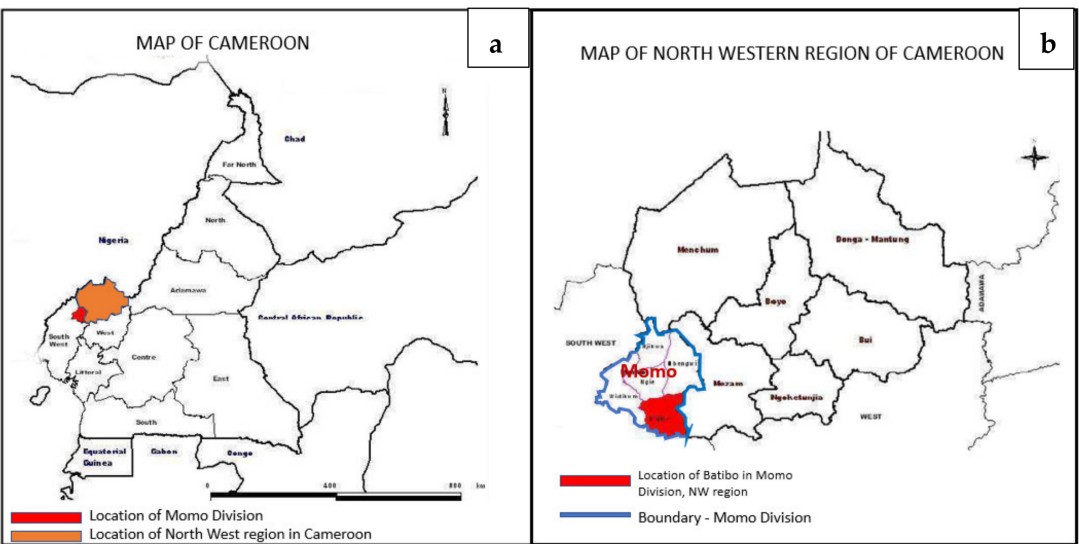

**Figure 5.** (**a**) Location of NW region in Cameroon; (**b**) Location of Batibo in Momo Division, NW region of Cameroon.

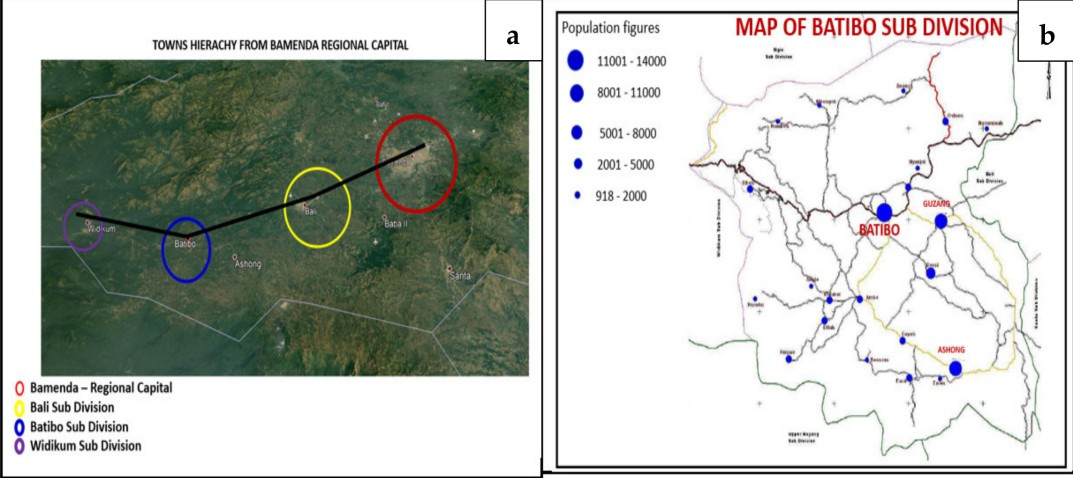

**Figure 6.** (**a**) Town sizes from Bamenda Regional Capital; (**b**) Population distribution of Batibo Municipality. Adapted with permission from Batibo Council. Batibo Council. Batibo Council Development Plan. PNDP 2012.

### 2.1. The 2012 Batibo Council Development Plan

Batibo municipality set as its vision in 2012 to be a modern municipality, disenclaved and sanitized, with a strong local economy and high-quality social wellbeing by 2020 [49]. The above vision was designed to be in line with the country's 2035 poverty alleviation strategy. To be able to emerge as a modern municipality by 2020, the council conducted a comprehensive community participation study of the 22 autonomous traditional villages that makes up the Batibo municipality to identify problems impacting the community's life quality and opportunities for development. Based on the study similar problems were identified in all the communities including Batibo town, which is the headquarter and center of the municipality. The problems include:

- No master plan or town planning for the communities;
- Erosion of indigenous cultural heritage;
- Unsustainable exploitation of the environment;
- High unemployment rate;
- Under exploitation and under development of touristic sites;
- Inadequate access to basic education;
- Inadequate access to public health;
- Inadequate access to water and electricity supply;
- Poor inter and intra village road network;
- Difficulty in marketing agriculture products;
- Low crop productivity;
- Women marginalization;
- Inadequate access to social services.

To address the above problems, a council development plan, the first ever development plan since the creation of the Batibo administrative unit in 1968 was developed with the help of a consultant, the government and some international organizations. The plan identifies development needs in each of the 22 villages that makes up the municipality. The needs were then prioritized at the municipal level for execution with cost estimates assigned and indicators to measure the impact of each project to the community were also provided. According to the plan, priority projects included investment in preserving and promoting culture, agriculture, environmental protection and awareness, social services, town planning and job training opportunities, youth and civic education, health care, public works, small and medium size enterprise, water and energy and digital communication network (Appendix A—Table A1).

### 2.2. Outcome of the 2012 Council Development Plan as of April 2021

Between 20 February and 15 April 2021, some relevant stakeholders in the community were contacted to talk about their views on the state of implementation of the 2012 development plan and vision of the municipality. A total of twenty interviews were conducted. The questions for the interview (Appendix A) were based on the list of initiatives identified by the municipality for implementation by 2020. The purpose of the interview was to identify initiatives that have been implemented, assess the impact to the community, examine the priority of implementation of the initiatives, assess if the 2020 vision was realized and, finally, determine the current status of the community in relation to sustainability development and circularity. The outcome from the interview is summarized in Table 1 below using broad heading of the various initiatives.

**Table 1.** Summary of interview responses.

| | Scale | | | | |
|---|---|---|---|---|---|
| **General Questions** | **%** | **%** | | **%** | **%** |
| | **Yes** | **No** | **High** | **Low** | **Moderate** |
| Awareness of the plan | 70 | 30 | | | |
| Participation in the plan design | 70 | 30 | | | |
| Expectations of the plan in 2012 | | | | 80 | 20 |
| Expectations realized | 5 | 95 | | | |
| Rate implementation of the plan | | | | 95 | 5 |
| Funding for projects received | 0 | 100 | | | |
| Continue with plan as it is or revised | 50 | 50 | | | |
| New plan required | 60 | 40 | | | |
| | Obeservation | | | | |
| **Batibo priority projects for its 2020 vision** | **Major change** | **Minor change** | **No change** | **% change** | **Status better/worse/ same** |
| Agriculture | | √ | | | SAME |
| Cultural development | | √ | | | SAME |
| Vocational training | | | √ | | SAME |
| Environment | | √ | | | SAME |
| Forestry and wildlife | | | √ | | SAME |
| Housing and urban development | | | √ | | SAME |
| Labor and social security | | | √ | | SAME |
| Livestock, fisheries and animal industry | | | √ | | SAME |
| Mines, industry and technological development | | | √ | | SAME |
| Public health | | | √ | | SAME |
| Public works | | | | | SAME |
| Basic education | | √ | | | BETTER |
| Social affairs | | | √ | | SAME |
| Sports and physical education | | | √ | | SAME |
| Property and land tenue | | | √ | | SAME |
| Tourism | | | √ | | SAME |
| Trade | | | √ | | SAME |
| Women empowerment | | √ | | | SAME |
| Youth employment | | | √ | | SAME |
| Business and economy | | | √ | | SAME |
| Water and energy | | | √ | | SAME |
| Communication | | | √ | | SAME |
| Higher education | | | √ | | SAME |
| Administrative and financial autonomy | | | √ | | SAME |

Table 1 shows that the majority of initiatives programmed by the municipality for implementation by 2020 were not realized. As a result, the respondents concluded that the community did not achieve the 2020 vision outlined in the 2012 strategic plan. As noted, some minor improvements were made in specific areas. The improvements identified by respondents include the grading of seasonal roads in some communities, planting of trees, new basic and secondary schools, agricultural extension programs, cultural programs and the creation of some health posts. As noted in Table 1, the improvements did not have any meaningful change on the quality of life in the community. It did not solve the jobs or economic problems, nor the rural exodus in the area. The reasons forwarded by the respondents for the failed outcome of the 2020 vision include the following:

- The ongoing arm conflict in the region that started in 2016 constituted a major factor hindering implementation of some of the plan's initiatives. As noted by the respondents, the cause of the conflict is partly due to the poor living condition in the area and poor governance under which the municipality had been operating;
- Another reason forwarded was the lack of decision-making autonomy in the management of council affairs. As noted by the respondents, elected council officials did not have full authority to make certain decisions or to redirect the focus of municipality resources to speed up development. They identified some schools and health posts financed by the government where the decision to locate the structures were chosen by the government even though the locations were not the ideal locations proposed by the local council authorities. They concluded that dependence on the central government for decision making slowed the realization of the vision;
- Funding was identified as major cause for the failure in realizing the vision. The respondents noted that the expected human and financial support from the government to finance the proposed projects were not received.

A survey also was conducted requiring the respondents to rank the initiatives proposed in the development plan in order of value to them and the community in general for implementation. The top three initiatives from each respondent were classified as high priorities, the next three were classified as medium and the last four were group as low priority initiatives (Table 2).

**Table 2.** Rankings of respondents from high, medium to low priorities. Rank the top 10 in order of importance: 1. Road construction; 2. Basic education; 3. Vocational training; 4. Water and electricity; 5. Public health; 6. Tourism and culture development; 7. Town plan for Batibo Municipality; 8. Mechanized agriculture; 9. Agricultural extension program; 10. Environmental preservation; 11. Social services; 12. Sports and recreation; 13. Market construction; 14. Telecommunication extension.

| Respondents | Rankings | | | | | | | | | |
|:---:|:---:|:---:|:---:|:---:|:---:|:---:|:---:|:---:|:---:|:---:|
| | High Priorities | | | Medium Priorities | | | Low Priorities | | | |
| A | 7 | 1 | 4 | 5 | 2 | 3 | 8 | 9 | 6 | 10 |
| B | 7 | 1 | 4 | 5 | 8 | 9 | 2 | 3 | 6 | 10 |
| C | 3 | 4 | 9 | 8 | 2 | 7 | 1 | 5 | 6 | 10 |
| D | 5 | 2 | 3 | 4 | 1 | 9 | 8 | 7 | 6 | 10 |
| E | 13 | 7 | 4 | 1 | 8 | 5 | 14 | 12 | 9 | 6 |
| F | 2 | 1 | 3 | 5 | 7 | 4 | 9 | 8 | 10 | 11 |
| G | 2 | 3 | 5 | 8 | 1 | 4 | 11 | 13 | 14 | 10 |
| H | 8 | 1 | 3 | 7 | 4 | 2 | 14 | 5 | 6 | 12 |
| I | 4 | 5 | 3 | 8 | 9 | 2 | 1 | 7 | 13 | 10 |
| J | 4 | 3 | 8 | 5 | 7 | 14 | 2 | 13 | 6 | 10 |

**Table 2.** *Cont.*

| Respondents | Rankings | | | | | | | | | |
| --- | --- | --- | --- | --- | --- | --- | --- | --- | --- | --- |
| | High Priorities | | | Medium Priorities | | | Low Priorities | | | |
| K | 4 | 1 | 7 | 3 | 5 | 2 | 14 | 8 | 9 | 13 |
| L | 1 | 2 | 3 | 5 | 4 | 7 | 9 | 10 | 8 | 10 |
| M | 1 | 9 | 4 | 8 | 3 | 6 | 5 | 7 | 2 | 10 |
| O | 4 | 7 | 3 | 5 | 6 | 8 | 9 | 2 | 13 | 11 |
| P | 1 | 7 | 8 | 4 | 9 | 2 | 3 | 6 | 14 | 11 |

In each group on Table 2, frequencies were calculated to identify their order of priority. Initiatives with lower frequencies in one group are compared with its frequency in the next lower group before it is finally ranked (Table 3).

**Table 3.** Frequency table of priority initiatives for Batibo municipality. Initiatives: 1. Road construction; 2. Basic education; 3. Vocational training; 4. Water and electricity; 5. Public health; 6. Tourism and culture development; 7. Town plan for Batibo Municipality; 8. Mechanized agriculture; 9. Agricultural extension program; 10. Environmental preservation; 11. Social services; 12. Sports and recreation; 13. Market construction; 14. Telecommunication extension.

| High Prorities | | Medium Prorities | | Low Priorities | |
| --- | --- | --- | --- | --- | --- |
| Initiative | Frequency | Initiative | Frequency | Initiative | Frequency |
| 1 | 8 | 1 | 7 | 1 | 2 |
| 2 | 4 | 2 | 5 | 2 | 3 |
| 3 | 4 | 3 | 3 | 3 | 1 |
| 4 | 9 | 4 | 4 | 4 | 0 |
| 5 | 2 | 5 | 7 | 5 | 3 |
| 6 | 0 | 6 | 2 | 6 | 7 |
| 7 | 5 | 7 | 5 | 7 | 3 |
| 8 | 1 | 8 | 7 | 8 | 8 |
| 9 | 2 | 9 | 3 | 9 | 5 |
| 10 | 0 | 10 | 0 | 10 | 11 |
| 11 | 0 | 11 | 0 | 11 | 3 |
| 12 | 0 | 12 | 0 | 12 | 2 |
| 13 | 1 | 13 | 0 | 13 | 3 |
| 14 | 0 | 14 | 1 | 14 | 1 |

| | High priority | | Medium priority | | Low priority |
| --- | --- | --- | --- | --- | --- |

As noticed in Table 3, the most needed initiatives for the community ranked by the respondents include roads, a town master plan, water and electricity and vocational training. The least needed investment initiatives are environmental, tourism and cultural projects. Basic education, agricultural extension programs are medium priority projects to the community thus investments in these sectors could have some impact but not compared to high value initiatives. Comparing the rankings of priorities in Table 1 to Table 3, a disparity is observed between initiatives that have been implemented and what is most wanted by the community.

*2.3. Assessment of the Plan in Relationship to Circularity*

The 2012 development plan of Batibo municipality was not developed with a circular agenda but most of the initiatives proposed in the plan are circular initiatives since they can promote sustainability and circularity if implemented. Based on the outcome identified and the baseline information provided by the plan, the community still is a community

with high poverty rate, poor quality of life, an unsustainable and a non-circular community. It also shows that the few initiatives implemented failed in revitalizing the community to the satisfaction of the community residents. Though the respondents provided reasons for the failure of the plan discussed above, from circular city perspective, the following reasons are attributed to the failure.

- The plan appears to be too broad. This makes it expensive and difficult to implement all the proposed initiatives within the time frame provided;
- The plan did not scale the initiatives or group the initiatives in order of value to identify those that would have the most economic, social, environmental and cultural impact in the community, improve the quality of life and catalyze development through the multiplier effect in the community;
- The time frame provided for the realization of the very broad plan was too short. It failed to capture the complexity involved in implementing certain projects;
- In setting the time frame for the vision, the plan failed to factor the enablers such as governance system and town planning that has been acting as legal barriers for development in rural areas.

### 2.4. Circular Approach for Revitalization of the Batibo Municipality

The Batibo development plan provided an extensive list of priority initiatives that can improve the quality of life and eradicate poverty in the community if implemented. Given the fact that investments have been made on some of the initiatives in the past decades, and during the period of the implementation of the development plan (2012–2020) and no change in the quality of life of the community is observed shows that identification and implementation of the appropriate solution is still a problem. Given the above challenge, the circular prioritization approach fits to this situation and is recommended for the community.

### 2.5. Circular Prioritization Approach

The relatively low impact on poverty levels, attractiveness and competitiveness identified after project implementations, in addition to the variation identified between what is needed by the community and what is implemented by the government signifies that low value initiatives are given more priority for implementation than high value initiatives. Using the circular prioritization chart, the community can arrange all the initiatives proposed in the strategic plan based on the value or impact they could have to the community and the ease of implementation. The initiatives or groups of initiatives with the highest value and considerable ease of implementation are given priority for execution. Figure 5 below shows possible arrangement of the various initiatives on the chart that resulted to the failed outcome of the 2020 vision and the continuous increase in poverty despite investments made over the past decade. As noticed, low value initiatives that are easy to implement are giving priority for implementation compared to the higher value initiatives identified on the survey. Using Table 3 survey results, Figure 7a below shows the current prioritization and Figure 7b below displays the possible arrangement of initiatives on the circular chart that if implemented could produce the desired outcome for poverty alleviation in the community.

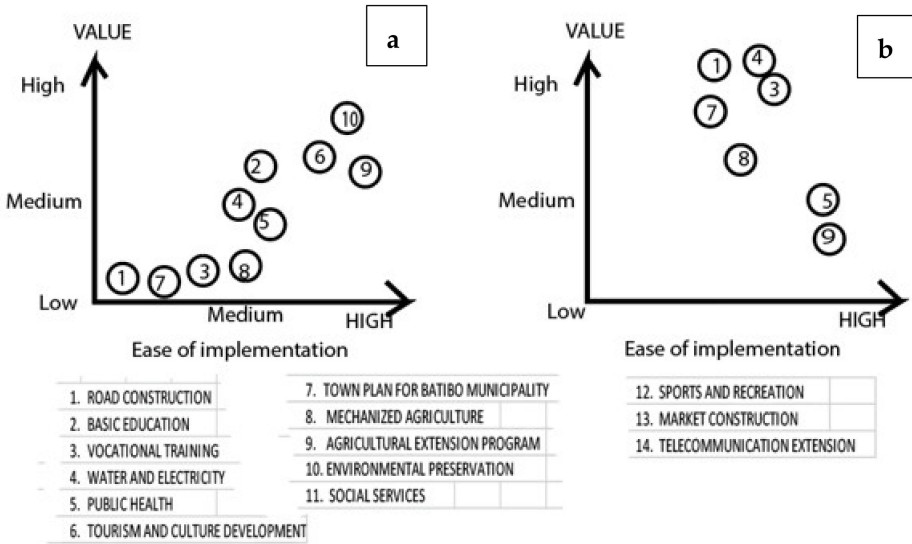

| | |
|---|---|
| 1. ROAD CONSTRUCTION | 7. TOWN PLAN FOR BATIBO MUNICIPALITY |
| 2. BASIC EDUCATION | 8. MECHANIZED AGRICULTURE |
| 3. VOCATIONAL TRAINING | 9. AGRICULTURAL EXTENSION PROGRAM |
| 4. WATER AND ELECTRICITY | 10. ENVIRONMENTAL PRESERVATION |
| 5. PUBLIC HEALTH | 11. SOCIAL SERVICES |
| 6. TOURISM AND CULTURE DEVELOPMENT | |

| | |
|---|---|
| 12. SPORTS AND RECREATION | |
| 13. MARKET CONSTRUCTION | |
| 14. TELECOMMUNICATION EXTENSION | |

**Figure 7.** (**a**) Current prioritization; (**b**) Proposed prioritization.

## 3. Results

*Conceptual Town Center Plan for Batibo Municipality*

The development of a town plan for the Batibo community is identified as a high value initiative in the survey above. To create a brand for socioeconomic, environmental and cultural growth for the community, the conceptual town center plan shown below is proposed (Figures 8–11). The plan involves a combination of high value and low value initiatives. The site selected for the project is the current site of the Batibo town hall (Figures 8 and 9), and the community field. The site is selected because of its accessibility and visibility from the main surfaced road that connects to new roadways in the community, and its historic nature as the center of sociocultural and economic activities in the area. The problems solved in this design include the following:

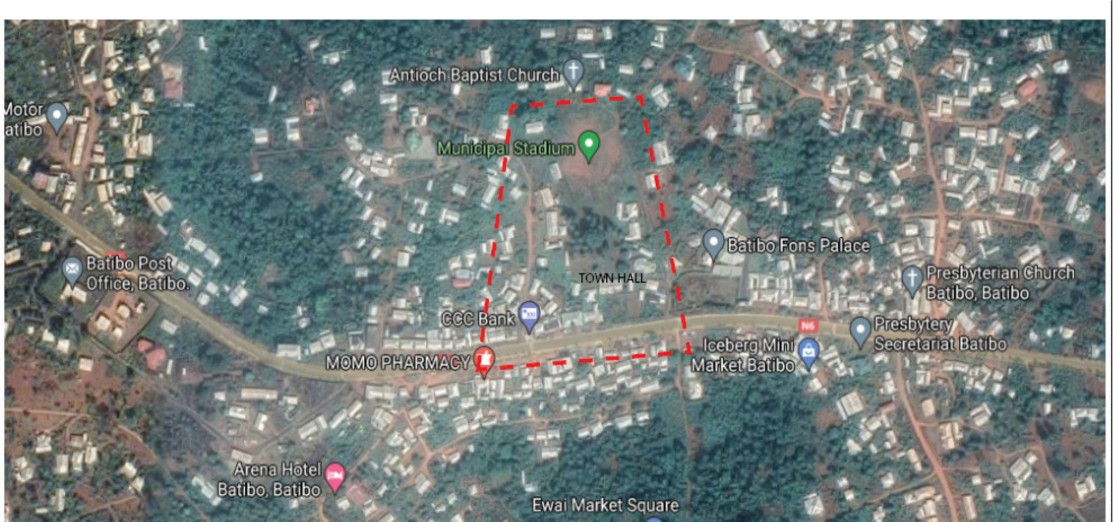

**Figure 8.** Google map aerial view of Batibo town showing the proposed site for the new town center.

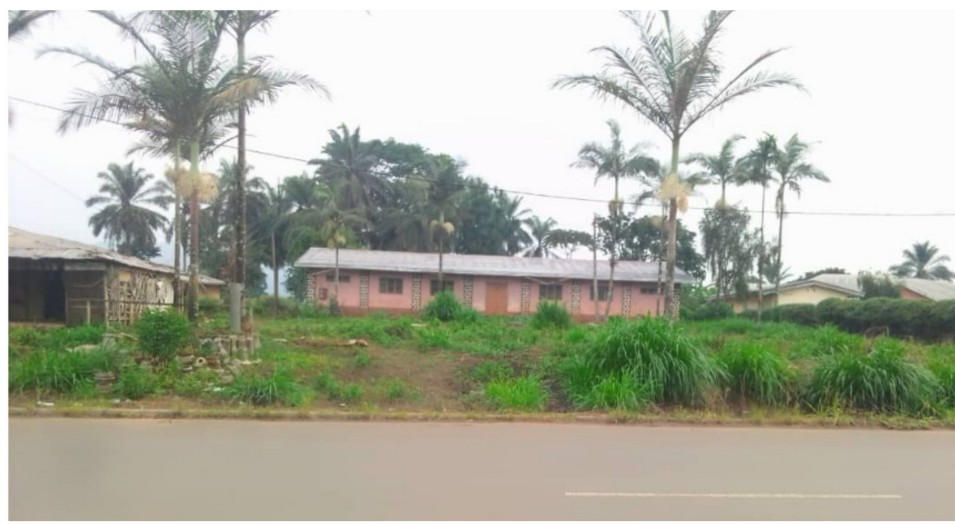

**Figure 9.** Existing Batibo town hall.

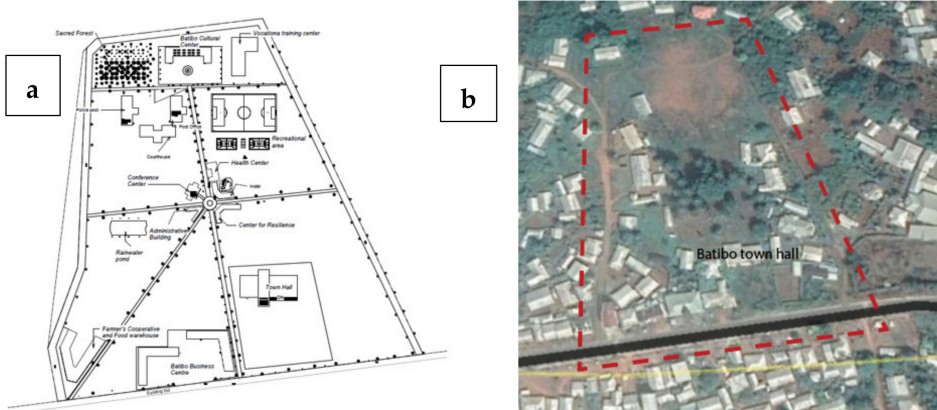

**Figure 10.** (**a**) Proposed Batibo town center; (**b**) Existing site conditions.

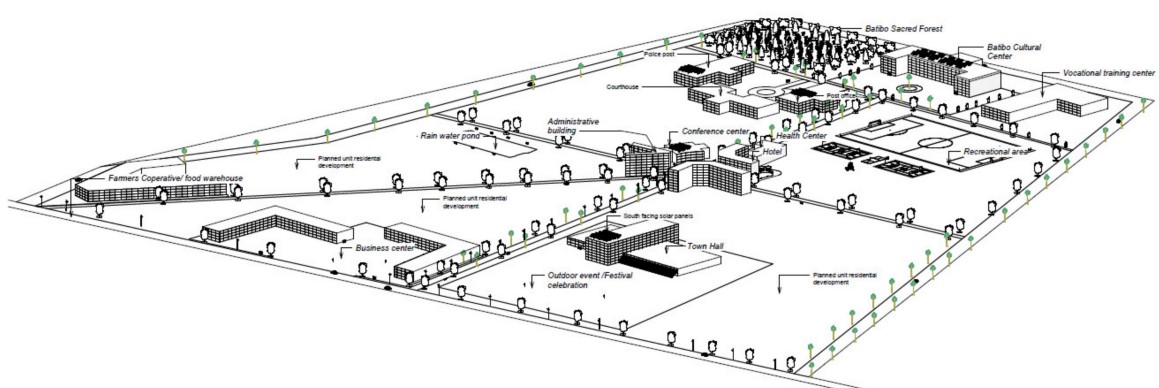

**Figure 11.** Proposed town center plan with recommended public facilities and open spaces for planned residential unit development.

Cultural heritage: The poor accessibility to some of the twenty-two traditional chiefdoms in the community hinders the ability for visitors and tourists to visit the sites hosting the cultural heritage. In addition, poor preservation and restoration in these remote locations exposes the heritage to destruction. This plan proposes an imposing architectural edifice—cultural history building—for a cultural center where the cultural heritages can be displayed for easy accessibility to tourists and other visitors, and for easy restoration

and preservation. Since dancing, traditional religious rites and palm wine production and drinking is typical to the culture of the people, the plan proposes a sacred forest next to the cultural center for performance of sacred traditional and religious rites. The streets within the site are aligned with palm trees and other native trees to celebrate the palm wine culture and proposes outdoor and indoor spaces for celebration of cultural festivals and other events. Spaces for a library, training and education on the cultural heritage is proposed in this center. A theater for arts performance is proposed on the site to encourage creative arts in the community.

Water and electrical infrastructure: The unreliable water and electricity supply in the community is another problem solved in this plan. The plan takes advantage of the abundant rainfall and sunlight in the area to provide rainwater storage ponds. Roof top solar panels and solar powered street lights are proposed to provide additional energy source for the community. In addition, the proposed buildings are designed with south facing windows to provide enough day light into the habitable spaces to reduce the use of electricity.

Vocational training: Vocational training center also is proposed to train community dwellers on waste recycling technics and other job skills. To encourage recycling and proper waste disposal recycling bins are provided at various locations on the site. To ensure proper collection of circular data in the community, ensure distribution and sharing of the data with business entities and other stakeholders, a center for innovation and resilience is proposed. The center is provided with spaces for training, conferences and information technology. Other facilities proposed on this site include the business center, warehouse for food and crop storage and quality outdoor recreational facilities to improve attractiveness. Public service institutions also are proposed to be located on this site. The proximity of the facilities is designed to encourage easy exchange and flow of resource within the community and to urban areas to strengthen markets.

## 4. Discussion

Governance systems and government policies impact the ability of both urban and rural communities to revitalize. The length of time it takes government institutions to figure out the impact of policies on community development, identify and implement the appropriate solutions affects the time it takes for communities to alleviate poverty and become sustainable and competitive. Removal of barriers for sustainable development as is the case of Cameroon does not guarantee quick revitalization and sustainability of the community. Quick revitalization is fostered by the ability of communities to identify problems, provide a comprehensive plan based on town planning principles and prioritize and implement initiatives or groups of initiatives that provide the highest value or impact to the community. Since communities may end up investing in low value initiatives if not properly informed of the choices and their relationship to the broader agenda of creating a competitive, attractive and sustainable community, vision and plan development is strengthened with education of the community. While agriculture is a good option for rural development and usually given high priority for investments in rural areas, this in addition to the associated basic services provision, cannot stop rural exodus, attract new visitors or residents and improve quality of life of the rural dwellers. Investments in the creation of planned, attractive and vibrant town centers to serve as market or increase market potential for the agricultural products, provide opportunities for recreation and jobs and provide quality environment for the display of the culture and identity of the community is equally needed since these are what attracts visitors and tourists. It should be noted that, the choice of town planning and the focus of a culturally led town center master plan development as an alternative approach to revitalize communities is not new. This approach has been time tested. The problems of gentrification and dubious long-term value of flagship projects identified by Gainza, X. [41] or the encroachment into agricultural land discussed by Harvold, Kjell and Berit Nordahl [32] are issues that can be properly addressed in the comprehensive planning process. They cannot be an excuse for not adopting an alternative

approach for the betterment of the community. Besides, adopting alternative approach does not mean an end to investment in agricultural activities or provision of basic services in other areas in the community. Expecting rural communities to alleviate poverty and become sustainable without proper planning of the built environment to ease the provision, circulation and distribution of goods and services, sets a very high bar for the already weak rural communities to attain.

In summary, this paper highlights the importance of governance and town planning in the development of a community. It emphasizes the significance of best practices that are essential to the development agenda for a sustainable community. As a top priority, a town center master plan is recommended in this study as a starting point for the revitalization and branding of the community. Issues and data collection to reveal factors that may limit its application and successful planning, such as land ownership, status of current infrastructure, available funding, market potential and professional expertise, are deserving of further study.

**Author Contributions:** Conceptualization, methodology, validation, investigations and formal analysis, M.M.; writing—original draft preparation, M.M.; conceptualization, methodology, writing—review and editing and supervision, A.F. All authors have read and agreed to the published version of the manuscript.

**Funding:** This research received no external funding.

**Institutional Review Board Statement:** The University of the District of Columbia oversight and approval, 21 April 2021.

**Data Availability Statement:** Not applicable.

**Conflicts of Interest:** The authors declare no conflict of interest.

### Appendix A. Interview Questions

**Research Topic:** Revitalization and Branding of Rural Communities in Cameroon using a Circular Approach for Sustainable Development—a Proposal for Batibo Municipality.

The purpose of the interview is to evaluate the implementation of the 2012 strategic development plan of the Batibo municipality. The questions here are related to the plan that was published and made available on the internet. The method used for this questionnaire is direct interview and survey. In participating in this interview, your identity or position shall not in any way be exposed or mentioned in the study. In addition, you can withdraw from the interview at any time or decide not to answer any question or questions. There is no benefit offered in participating in this study. Should you have any concern regarding the interview, please contact Dr. Anna Franz at 202-657-3794 or email anna.franz@udc.edu.

1. Are you aware of the Batibo Council Development plan published in 2012?

   YES NO

2. Were you or any member of your community involved in the study?

   YES NO

3. What can you say about the plan in terms of its degree of involvement of the community?

   GREATLY INVOLVED LESS INVOLVED NO INVOLVEMENT

4. Did you have the opportunity to review the published report?

   YES NO

5. If yes, did you think the plan will help improve living condition in Batibo?
6. In your opinion, what is the good part of the plan?
7. In your opinion, what is the bad part of the plan?
8. What would you have liked the plan to focus on?
9. Did you expected the plan to succeed in its objectives?

YES NO Explain

10. Now is 2021. has any project in the plan been wholly or partially implemented?

If yes, which part
If no, why not

11. From the list of initiatives proposed by the community, please indicate which of the projects have been fully or partially implemented.
12. What change have you observed since the start of the implementation of the plan:

MAJOR CHANGE MINOR CHANGE NO CHANGE

13. What change do you expect to see in order for the plan to be implemented?
14. Do you think the plan should be revised or it should continue as it is?
15. Now that the plan has or has not implemented, what strategy did you think will work better moving forward?
16. Of all the priorities listed, rank them in order of importance to you and the community for implementation?
17. What do you think should be done as a priority to attract people and businesses to Batibo?
18. If a plan to rebuild Batibo town center emerge, will you support it to be located on a new site or here at the Batibo park area which is already operating like Batibo town center?
19. What are some of the facilities you would like to see in a new Batibo town plan?
20. In your opinion, what is the greatest obstacle for development in Batibo municipality?
21. In terms of culture, what cultural aspect of the plan has experienced change.
22. What in the plan has been done to improve livability, attractiveness and competitiveness of the municipality?

**Table A1.** Questionnaire to evaluate implementation of the Batibo Council 2012 Strategic Plan.

| Questionnaire to Evaluate The Implementation of Batibo Council Strategic Plan between 2012–2020 | | | | | |
|---|---|---|---|---|---|
| | Outcomes | | | | |
| **Programs** | **Implemented** | **Change** | **% Change** | **No Change** | **Status Better/Worst/Same** |
| **Agriculture** | | | | | |
| % of women in agriculture program | | | | | |
| capacity building on modern livestock | | | | | |
| creation of functional farmer's group | | | | | |
| creation of farmer's credit union | | | | | |
| capacity of farmers on financial management | | | | | |
| mechanized agriculture | | | | | |
| access to quality technical service | | | | | |
| public partnership network | | | | | |
| organization of agricultural show | | | | | |
| trained 1000 farmers on agriculture Improvements | | | | | |
| warehouse creation | | | | | |
| purchase of heavy duty equipment | | | | | |
| creation of agriculture posts | | | | | |
| employment of agriculture technology | | | | | |
| oil palm plantation creation | | | | | |

**Table A1.** *Cont.*

| Questionnaire to Evaluate The Implementation of Batibo Council Strategic Plan between 2012–2020 | | | | | |
|---|---|---|---|---|---|
| | Outcomes | | | | |
| **Programs** | **Implemented** | **Change** | **% Change** | **No Change** | **Status Better/Worst/Same** |
| funding provision for the above projects | | | | | |
| **Cultural Development** | | | | | |
| cultural festivals | | | | | |
| cultural structures construction | | | | | |
| annual arts exhibition | | | | | |
| training on local dialect and instructors | | | | | |
| museum construction | | | | | |
| rehabilitation of village palaces and halls | | | | | |
| funding provision | | | | | |
| **Vocational Training** | | | | | |
| creation of training centers | | | | | |
| provision of training equipments | | | | | |
| lobbying by mayor for training centers | | | | | |
| funding provision | | | | | |
| **Environment** | | | | | |
| natural resource map provision | | | | | |
| sanitation program installation | | | | | |
| recruitment & training sanitary workers | | | | | |
| provision of gabbage cans | | | | | |
| construction of public toilets | | | | | |
| street decorations and public places | | | | | |
| land use map provision | | | | | |
| gabbage trucks purchase | | | | | |
| landuse training of pop. | | | | | |
| provision of landfill site | | | | | |
| planting of ornamental trees | | | | | |
| creation of green spaces | | | | | |
| funding availability | | | | | |
| **Forestry and Wildlife** | | | | | |
| practice of conventional forest activities | | | | | |
| enforcement of illegal hunting | | | | | |
| sensitization on climate change activities | | | | | |
| game farms creation | | | | | |
| replacement of deforested areas | | | | | |
| inventory of tree species | | | | | |
| **Housing and Urban Development** | | | | | |
| development of master plan | | | | | |
| provision of basic utilities | | | | | |
| % town planning certificate issueance | | | | | |
| road maintenance | | | | | |
| water and electricity provision | | | | | |

**Table A1.** *Cont.*

| Questionnaire to Evaluate The Implementation of Batibo Council Strategic Plan between 2012–2020 | | | | | |
|---|---|---|---|---|---|
| | Outcomes | | | | |
| Programs | Implemented | Change | % Change | No Change | Status Better/Worst/Same |
| identification of non-conforming housing | | | | | |
| % of surfaced roads | | | | | |
| feasibility studies on roads | | | | | |
| funding availlability | | | | | |
| **Labor and Social Security** | | | | | |
| % of citizens with social insurance | | | | | |
| pension provision inplace | | | | | |
| labor security | | | | | |
| service providers | | | | | |
| employment contract implementation | | | | | |
| **Livestock, Fisheries and Animal Industry** | | | | | |
| increase in household income of farmers | | | | | |
| beneficiaries of assistance | | | | | |
| modern livestock farming | | | | | |
| fish farming creation | | | | | |
| change in farmers/grazier conflict | | | | | |
| dairy processing creation | | | | | |
| slaughter house creation | | | | | |
| training of farmers and training opportunities | | | | | |
| range creation | | | | | |
| veterinary service creatiion | | | | | |
| livestock cooperative creation | | | | | |
| cooperative finance training | | | | | |
| funding provision | | | | | |
| **Mines, industry and technological development** | | | | | |
| exploitation of new mining sites | | | | | |
| number of small scale industries | | | | | |
| authorization of operation of small scale industries | | | | | |
| funding provision | | | | | |
| **Public health** | | | | | |
| number of health care centers created | | | | | |
| proximity to health centers | | | | | |
| % of health care accessibility | | | | | |
| death rates under 5 years | | | | | |
| rehabilitation of healthcare centers | | | | | |
| provision of water points | | | | | |
| number of waste disposal system constructed | | | | | |
| recruitment of health staff | | | | | |
| funding provision | | | | | |

**Table A1.** *Cont.*

| Questionnaire to Evaluate The Implementation of Batibo Council Strategic Plan between 2012–2020 | | | | | |
|---|---|---|---|---|---|
| | Outcomes | | | | |
| **Programs** | **Implemented** | **Change** | **% Change** | **No Change** | **Status Better/Worst/Same** |
| **Public works** | | | | | |
| number of road projects implemented | | | | | |
| funding provision | | | | | |
| **Basic education** | | | | | |
| new secondary and technical college creation | | | | | |
| construction of infrastructure | | | | | |
| student teacher ratio | | | | | |
| lobby for recruitment of teachers | | | | | |
| funding and resource provision | | | | | |
| **Social Affairs** | | | | | |
| creation of social centers | | | | | |
| income generation activities for vulnerable persons | | | | | |
| funding provision | | | | | |
| **Sports and Physical Education** | | | | | |
| hosting of nation and international sporting activities | | | | | |
| creation of sports complex | | | | | |
| acquisition of land for sports complex | | | | | |
| funding provision | | | | | |
| **Property and Land Tenue** | | | | | |
| creation of cadastral map | | | | | |
| land title ownership | | | | | |
| land registration | | | | | |
| demarcation and registration of council land | | | | | |
| **Tourism** | | | | | |
| identification and development of touristic sites | | | | | |
| statistics on tourist visits | | | | | |
| creation of tourism board | | | | | |
| order from DO for local tourism board | | | | | |
| promotional activities | | | | | |
| funding provision | | | | | |
| **Trade** | | | | | |
| implementation of trade laws (%) | | | | | |
| security measures | | | | | |
| trade shows organized | | | | | |
| new market constructed | | | | | |
| rehabilitation of market | | | | | |
| funding provision | | | | | |
| **Women Empowerment** | | | | | |
| creation of women centers | | | | | |

**Table A1.** *Cont.*

| Questionnaire to Evaluate The Implementation of Batibo Council Strategic Plan between 2012–2020 | | | | | |
|---|---|---|---|---|---|
| | Outcomes | | | | |
| Programs | Implemented | Change | % Change | No Change | Status Better/Worst/Same |
| recruitment of training staff | | | | | |
| women's exhibition | | | | | |
| funding provision | | | | | |
| **Youth Employment** | | | | | |
| proportion of youths benefiting from employment programs | | | | | |
| funding provisions | | | | | |
| **Business and Economy** | | | | | |
| change in percentage of tax payers | | | | | |
| access to credits | | | | | |
| no. of micro enterpreneurs | | | | | |
| change in business activities | | | | | |
| construction of handicraft center | | | | | |
| financial institutions availability | | | | | |
| funding provision | | | | | |
| **Water and Energy** | | | | | |
| access to permanent energy supply | | | | | |
| access to drinking water | | | | | |
| water management committees (WMC) | | | | | |
| water systems availability | | | | | |
| extension and availability of electricity supply | | | | | |
| number of wells created | | | | | |
| water catchment protection | | | | | |
| extension of gravity water schemes | | | | | |
| training of WMC | | | | | |
| Funding provision | | | | | |
| **Communication** | | | | | |
| extension of communication network coverage | | | | | |
| local TV and radio channels creation | | | | | |
| funding provisions | | | | | |
| **Higher Education** | | | | | |
| number of higher institutions created | | | | | |
| funding provision | | | | | |
| **Territorial adm. and decentralization** | | | | | |
| recruitment of skilled staff | | | | | |
| capacity training | | | | | |
| funding provision | | | | | |

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
