# Peer review of "Revitalization and Branding of Rural Communities in Cameroon Using a Circular Approach for Sustainable Development—A Proposal for the Batibo Municipality"

_sustainability, doi:10.3390/su13126908_

Round 1

Reviewer 1 Report

The paper proposes a development program based on the circular economy in a country subject to a monetary regime such as the CFA franc, which is very problematic for mobilizing resources towards the effective development of these countries. To avoid discussing this difficulty, the authors quote D'Almeida (2019) in lines 204-206, "that circular city data is an effort to build a safe environment whereby start-ups, city agencies, and larger firms can collect, produce , access and exchange data, as well as business insights, through transaction mechanisms that do not necessarily require currency". However, this fact cannot be ignored. The governance over currency and finance is essential to direct development based on community and national interests in a monetary economy. See, for example, the work "Africa's Last Colonial Currency: The CFA Franc Story" by Fanny Pigeaud and Ndongo Samba Sylla (2021).
In addition, despite emphasizing community development, the culture-led development strategy is associated with attracting tourism, a trickle-down strategy. This type of strategy has serious distributional problems and triggers inflation before reaching the bottom. below, it must be added the armed conflicts in the area such as the one that occurred in 2016 that discourage tourism, as well as the current pandemic.
Furthermore, it is difficult to see how a strategy focused on attracting foreign investment and tourists can simultaneously reinforce local identity. The text avoids talking about the problems associated with this type of strategy, which can even expel its neighbors from revitalized areas. For example, see the article Culture-led neighborhood transformations beyond the revitalization / gentrification dichotomy by Xabier Gainza (2014). Building on Susan O'Hara's 5 Pillars of Development requires addressing the problematic issues mentioned in this review, and explaining how to make effective investments from a bottom-up approach, something that is evident given the lack of results found by the authors in the Batibo Council Development Plan of 2012.

Author Response

Responses:

CFA franc monetary issue: This paper fully considers the monetary issues plaquing the CFA region. However, this is not the focus of the paper. It is noted that despite the monetary challenges, investments still continue using the available resources. It is noted that investments by the government have been increasing in the past decades but its impact on economic growth and quality of life in the rural areas have not been noticeable (see Thierry, B. Jun, Z. and Guohu, H. 2007). This paper is forwarding that if this investment is accompanied with circular approach for project selection, prioritization and implementation, the result could be different. In addition, the fact that these rural areas lack a comprehensive master plan of their own to guide development means investment is not coordinated and defined to solve local problems that could improve quality of life.

Arms conflict, current pandemic and tourism: We agree that the arms conflict and the current pandemic are major obstacle for tourism. This paper points out that the poor life quality in the rural areas is a contributing factor to the conflict. The proposals in this paper are not short-term proposals. They are proposals that look to the future of the community. Applying these proposals could help prevent future conflicts. The paper considers that the conflict and the pandemic would end at some point and tourist activities would resume. The community that adopts the proposals contained in the paper improve the potential to boost tourism in their sector.

Culture led development strategy and distributional issue: Rural areas in Cameroon are “depressed areas” as described by Ndenecho [16] in the paper. This paper is looking at alternative approaches to solve the poor life quality in these areas since the existing approach is not working. This paper acknowledges the trickle-down challenges posed by this approach in addition to the other problems of gentrification and loss of agricultural land but recognizes that the benefit for adopting this approach outweighs the continuation of the existing approach. In addition, the approach is not new. It has been time tested and the problems identified above can be addressed in a comprehensive planning process.

Foreign investment and tourists and local identity issues: Attracting foreign investment is not the focus of this paper. The paper focuses on how to better invest the available resources to make the rural areas are attractive, competitive and economically sustainable. While foreign investment is something to consider, at this stage, the efficient use of the available local resources for development is the focus.

This paper agrees with Xabier Gainza that culture has been used and can be used as a means for revitalizing neighborhoods and branding the urban economy. The paper recognizes the downsides of such initiatives on the culture of the community. The paper concludes that a comprehensive town planning process can address the above downsides identified by Xabier in order to protect the culture and identity of the people. Giving that the development of this plan would be the first of its kind in the community, the availability of extensive examples and literature on this approach provides a good base for the community to expand on protect the culture and agricultural land.

Reviewer 2 Report

Dear authors,

The research needs a clarification of its main focus, conceptural framework, methodology, in sum, a total clarificaion is needed. The purpose is not clear, the context (rural, urban, town) is not clear. It is very confuse.

The decentralization process and the governance framewor is always a central aspect of territory development. However, as it is, we can not understand the exact terms of this process.

Regards.

Author Response

Responses:

Purpose of the study: The purpose of this paper is to demonstrate that circular enabling initiatives such as governance and town planning and their application process influences the ability of a community to revitalize, and become attractive and competitive. (blue highlight in revised paper)

rural, urban, town:  We agree that the term urban, city, rural are misleading. In the context of planning, the terms have been used to mean the same practice. The American planning Association (APA) used the term planning or city and urban planning interchangeably. The Royal Town Planning Institute in UK used the term town planning. Other terms such as country planning and regional and rural planning is used. The terms all refer to a field that tries to make towns, cities and the countryside attractive, safe and environment friendly. In this study, the author agrees and follows Carmona Matthew et al 2003 to conclude that the term urban, city, rural, town in this study have the same meaning.

Reviewer 3 Report

Starting from the local planning and investment opportunities generated by the recent Cameroon’s decentralization law, the authors devise directions for sustainable development in Batibo, a rural community in Cameroon. The manuscript is well-written, the literature review is extensive, the arguments are effective and clearly formulated.

Some issues to be considered:

  • While the ideas advanced by the authors are valuable, a critical question remains unanswered: when money is scarce, rising signature buildings in the town center can provide similar (if not bigger) returns compared to investing in infrastructure and education? Why should decision makers prioritize the proposed approach against more traditional alternatives?
  • It is a lengthy paper, but the final discussion section is too short. It could be a conclusion section.
  • Conclusions, limits and directions for future the research are missing.
  • The rather long section 1.1. “Legal History and 2019 Decentralization Law” is of limited interest for readers outside Cameron and should be downsized.

Author Response

Responses:

  • Lengthy paper, we agree. The intent is to explain all the concepts used in the paper. Circularity for example is a very recent concept and key word in this paper. Review of this in the paper is lengthy because we want to make it as understandable as possible for the readers.
  • Section 1.1, The section has been downsized. The last paragraph listing the news authorities of the local councils have been removed. This paper recognizes that many developing countries face the same issues.  It is critical to outline the complexity of changes and the time it takes to get move from an idea to decentralize until final authority. Tracing the history is important not only for understanding the challenges faced by the community and reasons for the lack of development, but also for other researchers to understand the importance of reviewing the details of governing policies and regulations.
  • The final discussion section is too short: The discussion part has been revised to recognize the challenges facing this approach.
  • Money scarcity and raising signature building: This paper considered this view but concluded that the available resources are not being used efficiently for the benefit of the community. This inefficiency increases cost and impacts the quality of life of the community. With the availability of local materials for construction, and with the help of professional architects, the issue of costs can be accommodated. In addition, this approach is not new. it has been time tested. The paper recognizes the issues of gentrification, destruction of arable land and long-term value of such projects associated with this approach but concludes that the benefits outweigh the cost if properly planned and executed.
  • Conclusions, limits and directions for future the research are missing: A paragraph was added at the end of the discussion to identify other important factors that are limiting and deserving of further study.

Round 2

Reviewer 1 Report

The changes made and the responses to the suggestions made are well justified. Although I believe that there is still a gap with respect to the CFA franc, the approach of the authors can resist the main criticisms to mobilize real resources

Reviewer 2 Report

Dear authors,

The paper was improved in accordance with the notes made.